# RsGCN: Subgraph-Based Rescaling Enhances Generalization of GCNs for Solving Traveling Salesman Problems

## Abstract

GCN-based traveling salesman problem (TSP) solvers face two critical challenges: poor cross-scale generalization for TSPs and high training costs. To address these challenges, we propose a Subgraph-Based Rescaling Graph Convolutional Network (RsGCN). Focusing on the scale-dependent features (i.e., features varied with problem scales) related to nodes and edges, we design the subgraph-based rescaling to normalize edge lengths of subgraphs. Under a unified subgraph perspective, RsGCN can efficiently learn scale-generalizable representations from small-scale TSPs at low cost. To exploit and assess the heatmaps generated by RsGCN, we design a Reconstruction-Based Search (RBS), in which a reconstruction process based on adaptive weight is incorporated to help avoid local optima. Based on a combined architecture of RsGCN and RBS, our solver achieves remarkable generalization and low training cost: **with only 3 epochs of training on a mixed-scale dataset containing instances with up to 100 nodes, it can be generalized successfully to 10K-node instances without any fine-tuning**. Extensive experiments demonstrate our advanced performance across uniform-distribution instances of 9 different scales from 20 to 10K nodes and 78 real-world instances from TSPLIB, while requiring **the fewest learnable parameters and training epochs** among neural competitors.

## 1 Introduction

**Background** The traveling salesman problem (TSP), as a typical combinatorial optimization problem, has been extensively studied in the literature. Although classical solvers like Concorde (Applegate et al., 2006) and LKH (Helsgaun, 2000) achieve strong performance across various problem scales, their designs highly rely on expert-crafted heuristics. The development of deep learning has spawned numerous neural TSP solvers with various learning paradigms (e.g., Supervised Learning, Reinforcement Learning) and architectural foundations (e.g., Graph Convolutional Networks (GCNs) (Joshi et al., 2019; Fu et al., 2021; Hudson et al., 2022; Sun & Yang, 2023; Li et al., 2023; 2024) and Transformers (Kool et al., 2019; Kwon et al., 2020; Drakulic et al., 2023; Luo et al., 2023; Pan et al., 2023; Ye et al., 2024; Li et al., 2025)). Herein, a typical framework is using neural networks to generate heatmaps and then incorporate post-search algorithms to obtain solutions with the guidance of heatmaps. LKH-3 (Helsgaun, 2017) and Monte Carlo Tree Search (Fu et al., 2021) are commonly employed in the post-search guided by heatmaps.

**Challenges** Although prior work has shown substantial potential of GCNs in solving TSPs, it also reveals critical limitations in solution generalization (Joshi et al., 2021). Specifically, pre-trained models cannot perform well for new problem scales, and thus further fine-tuning with new-scale instances is necessary to transfer pre-trained models to the new problem scales. Existing approaches to enhance generalization include: (1) enhancing solution diversity by generating multiple heatmaps (Li et al., 2023; 2024; Sun & Yang, 2023) or designing specific post-search (Drakulic et al., 2023; Fu et al., 2021; Hottung et al., 2022; Hudson et al., 2022), and (2) decomposing large-scale problems into small-scale sub-problems for partitioned solving (Fu et al., 2021; Li et al., 2025; Luo et al., 2023; Pan et al., 2023; Ye et al., 2024). Although these approaches achieve modest improvements, they suffer from prohibitive computational overhead. Regarding (1), generating additional heatmaps is

computationally intensive for GPUs, significantly diminishing the practicality. Furthermore, existing post-search algorithms exhibit insufficient capability in narrowing down the search space and thus result in limited search efficiency. As for (2), the decomposition of large-scale problems leads to a partial loss of global perspective and thus weakens the overall optimization effectiveness, while the design of sub-problems partitioning and recombination introduces additional complexity.

**Our Contributions**     Based on our research analysis, since GCNs are sensitive to numerical values, the scale-dependent features related to nodes and edges in TSPs are essential factors impairing generalization. In this paper, we propose a new GCN termed RsGCN, which incorporates the subgraph-based rescaling to adaptively rescale subgraph edges. As shown in Figure 1, the subgraph-based rescaling first (1) construct subgraphs for each node by $k$-Nearest Neighbor ($k$-NN) selection to ensure each node has a uniform number of neighbors regardless of instance scale; then (2) rescales subgraph edges by Uniform Unit Square Projection to maintain consistency of numerical distributions. With the subgraph-based rescaling, RsGCN can learn universal patterns by training on only small-scale instances. Thus, RsGCN can be effortlessly generalized to large-scale TSPs and generate high-quality heatmaps to guide post-search more efficiently. In addition, rethinking the limitations of existing post-search algorithms, we design a new post-search algorithm termed Reconstruction-Based Search (RBS), which utilizes 2-Opt as the basic optimizer and performs reconstruction based on adaptive weights to robustly escape from local optima. According to extensive experimental studies, the integrated framework of RsGCN and RBS shown in Figure 1 achieves **state-of-the-art** performance on both uniform-distribution and real-world instances among neural competitors.

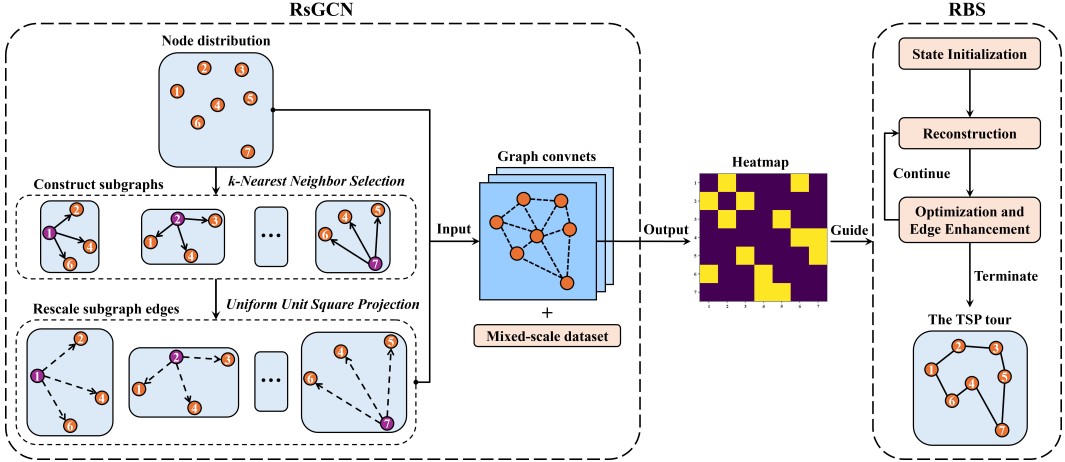

Figure 1: The integrated framework of RsGCN and RBS.

## 2 RELATED WORK

According to the model backbones, neural TSP solvers can be broadly categorized into Transformer-based and GCN-based methods. In this work, we focus on the generalization and cost of GCN-based methods.

GCN-based methods typically employ an edge-aware variant of GCN, namely Gated GCN (Bresson & Laurent, 2018). For GCN-based methods (Joshi et al., 2019; Fu et al., 2021; Xin et al., 2021; Hudson et al., 2022; Sun & Yang, 2023; Li et al., 2024), the trained GCN outputs informative heatmaps that indicate promising candidates. Subsequently, heatmaps are used to guide the post-search algorithm to construct the TSP tour. Much of the related research has centered on improving the generalization performance of GCNs. Joshi et al. (2021); Xin et al. (2021); Lischka et al. (2024) demonstrated that graph sparsification based on $k$-NN can improve the generalization of GCNs to some extent. Fu et al. (2021) proposed decomposing large-scale problems into multiple smaller sub-problems for separate solving, and then merging them using Monte Carlo Tree Search (MCTS). Moreover, Sun & Yang (2023); Li et al. (2024) employed diffusion models as decoders to generate diverse heatmaps, thereby enhancing solution diversity.

However, the decomposition-merging paradigm lacks a global perspective, leading to degraded solution quality. Incorporating Diffusion models as decoders introduces additional computational overhead while offering no substantive improvement to the graph encoder. Overall, prior work has overlooked the high sensitivity of edge-aware GCNs to edge weights, which is the essential factor limiting their generalization. Notably, a Transformer-based method, INViT (Fang et al., 2024), introduces invariant nested views to normalize node distributions, thereby enhancing the generalization for cross-distribution TSP. Building on prior studies and analyses, we propose a subgraph-based edge normalization method and design RsGCN, which incorporates a global perspective and strong cross-scale generalization. We empirically demonstrate that RsGCN can effectively generalize from small-scale to large-scale TSPs with very few parameters and low training cost.

## 3 PRELIMINARIES

**Problem Definition**  In symmetric TSPs, nodes are located in a 2-dimensional Euclidean space with undirected edges. For an $n$-node instance, $\mathcal{X}_n = \{x_1, x_2, \ldots, x_n\}$ denotes its set of node coordinates, where $x_i = (a_i, b_i)$. A solution for TSPs, i.e., a TSP tour, is a closed loop that traverses all nodes exactly once and is typically encoded by a permutation $\Pi_n = \{\pi_1, \pi_2, \ldots, \pi_n\}$. The length of the tour $\Pi_n$, denoted by $L(\Pi_n)$, is calculated as in Equation 1, where $\|\cdot\|$ denotes the Euclidean norm. Finally, the optimization objective is to find the permutation that minimizes the tour length.

$$L(\Pi_n) = \sum_{i=1}^{n-1} (\|x_{\pi_i} - x_{\pi_{i+1}}\|) + \|x_{\pi_n} - x_{\pi_1}\|. \tag{1}$$

**Heatmap Representation**  GCNs output a heatmap $\boldsymbol{H} \in \mathbb{R}^{n \times n}$ with input of an $n$-node instance, where $\boldsymbol{H}_{i,j}$ represents the heat of the edge $x_j \to x_i$, i.e., the probability that $x_j$ is the successor of $x_i$ in the predicted TSP tour. Indicative heatmaps can guide the post-search process, narrowing the search space and thereby improving solution quality.

## 4 METHODOLOGY

### 4.1 SUBGRAPH-BASED RESCALING FOR GCN – RSGCN

GCNs have powerful feature representation capability to generate high-quality heatmaps for fixed-scale TSPs. However, previous studies (Joshi et al., 2021; Sun & Yang, 2023) have shown that GCNs exhibit limited cross-scale generalization capability. Specifically, scale-dependent features vary in different scales of TSPs and thus weaken the generalization capability. To learn universal patterns and enhance cross-scale generalization, our RsGCN incorporates the subgraph-based rescaling to rescale subgraph edges, which consists of two steps as follows.

### 4.1.1 CONSTRUCT SUBGRAPHS

The connections between nodes in TSP form fully connected graphs, implying that the adjacent nodes of each node increase as the total number of nodes increases. This not only leads to a significant increase in computational complexity but also poses great challenges to the robustness of GCNs' representations when performing message aggregation on dense graphs of various scales.

To address this issue, we employ $k$-NN selection to prune the number of adjacent nodes as $k_1$ for each node, thereby each node forms a subgraph including itself and its top $k_1$ nearest neighbors. $k_1$ is fixed and thus the nodes in TSPs with various scales have subgraphs with a uniform number of adjacent nodes, helping learn universal patterns from instances with different scales. In addition, it can also achieve graph sparsification (Joshi et al., 2021; Xin et al., 2021; Lischka et al., 2024) that enhances GCNs by retaining promising candidates and stabilizing message aggregation.

Specifically, for an $n$-node instance with node set $\mathcal{X}_n = \{x_1, x_2, \ldots, x_n\}$, the subgraph set is denoted by $\boldsymbol{\mathcal{N}}(\mathcal{X}_n | k_1) = \{\mathcal{N}_1^{k_1}, \mathcal{N}_2^{k_1}, \ldots, \mathcal{N}_n^{k_1}\}$, where $\mathcal{N}_i^{k_1} = \{x_i^1, x_i^2, \ldots, x_i^{k_1}\}$ contains the $k_1$-nearest neighbors of $x_i$ (including $x_i$ itself) and represents $x_i$'s subgraph. $x_i^j = (a_i^j, b_i^j) \in \mathcal{X}_n$ denotes the coordinate of $x_i$'s $j$-th nearest neighbor.

### 4.1.2 RESCALE SUBGRAPH EDGES

In existing research, the coordinates of nodes in TSPs are typically normalized in a unit square space $[0,1]^2$. However, with the same range of space, node distributions tend to become denser as the problem scale increases, which makes edge lengths smaller on average. Therefore, cross-scale differences in edge lengths can impair the cross-scale generalization of GCNs.

To deal with the above issue, we introduce Uniform Unit Square Projection to rescaling subgraph edges. For a subgraph $\mathcal{N}_i^{k_1}$, the main idea is to project its node coordinates so that the coverage of $\mathcal{N}_i^{k_1}$ is maximized in the unit square space. In detail, we first acquire the maximum and minimum coordinate values on the two axes from $\mathcal{N}_i^{k_1}$, denoted by $a_i^{\max}$, $a_i^{\min}$, $b_i^{\max}$, and $b_i^{\min}$, respectively. Then, we compute the scaling factor $\mu_i$ and rescale subgraph edges according to Equations 2 and 3.

$$\mu_i = \frac{1}{\max(a_i^{\max} - a_i^{\min}, b_i^{\max} - b_i^{\min})} \tag{2}$$

$$\text{Rescale}(\|x_i - x_i^j\|) = \|x_i - x_i^j\| \cdot \mu_i, \ \forall j \in \{1, 2, \dots, k_1\}. \tag{3}$$

After rescaling the edges of all subgraphs in $\mathcal{N}(\mathcal{X}_n|k_1)$, the cross-scale differences in edge lengths are eliminated. For intuitive comprehension, an example of the subgraph-based rescaling is shown in Figure 2. The leftmost figure shows the original node distribution, while the red rectangle denotes the subgraph frame. The rightmost figure presents the rescaled subgraph. In addition, the effect of the subgraph-based rescaling is visualized and discussed concretely in Appendix G.

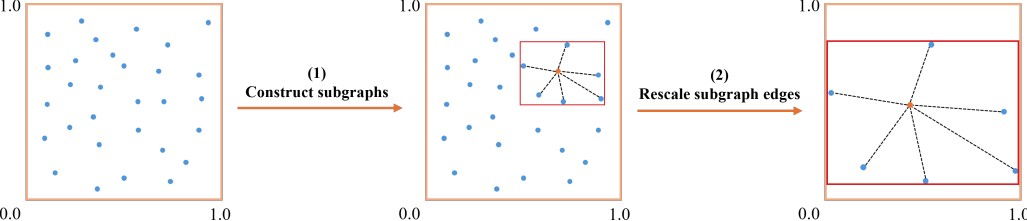

Figure 2: An example of the Uniform Unit Square Projection for a subgraph.

### 4.1.3 ARCHITECTURE OF GCN

The global node coordinates and the rescaled subgraph edge lengths serve as inputs to RsGCN, providing both global and local perspectives. Due to space limitations, this section has been placed in Appendix B.

### 4.2 TRAINING STRATEGY

**Mixed-Scale Dataset** A mixed-scale dataset is used to train RsGCN. In detail, the mixed-scale dataset includes one million TSP instances, in which the number of nodes varies in 20, 30, 50, and 100, and the proportion of these four scales is 1:2:3:4. On the one hand, the mixed-scale dataset effectively helps RsGCN learn universal patterns across different scales, avoiding the pitfalls of narrow patterns specific to a single scale. On the other hand, the maximum scale in the mixed-scale dataset is only 100 nodes, and RsGCN can be efficiently generalized to 10K-node instances without any fine-tuning, which can significantly reduce the training cost.

**Bidirectional Loss** According to the undirected nature of the symmetric TSP, we employ a double-label binary cross-entropy as the loss function. In detail, the two adjacent nodes of $x_i$ in the optimal tour $\hat{\Pi}$ serve as the two labels. The loss function $\mathcal{L}$ for $n$-node instances is presented as follows:

$$\mathcal{L} = -\frac{\sum_{x_i \in \mathcal{X}_n} \sum_{x_j \in \mathcal{X}_n} \mathcal{Z}_{i,j}}{n}, \tag{4}$$

$$\mathcal{Z}_{i,j} = \begin{cases} \log(\boldsymbol{H}_{i,j} + \varepsilon), & \text{if } x_j \text{ is adjacent to } x_i \text{ in } \hat{\Pi}, \\ \log(1 - \boldsymbol{H}_{i,j} + \varepsilon), & \text{others}, \end{cases} \tag{5}$$

where the division by $n$ in Equation 4 aims to balance the loss on different scales, helping prevent the gradient updates from being biased towards larger-scale instances. In Equation 5, $\boldsymbol{H}_{i,j}$ represents the sigmoid-transformed probability of the edge $x_i \rightarrow x_j$. $\varepsilon$ is a very small value used to avoid taking the logarithm of zero.

### 4.3 RECONSTRUCTION-BASED SEARCH – RBS

Inspired by the $k$-Opt operations in MCTS (Fu et al., 2021), we propose a post-search algorithm termed Reconstruction-Based Search (RBS). RBS employs a candidate-based search strategy, enabling its search results to clearly reflect the quality of the heatmaps, thereby facilitating the evaluation of the generalization ability of GCNs. The overall procedure of RBS is shown in Figure 1, which first conducts **State Initialization** to obtain an initial tour and then iteratively conducts **Reconstruction** and **Optimization and Edge Enhancement** for further optimization until the time budget expires.

#### 4.3.1 STATE INITIALIZATION

Let $k_2$ denote the size of the candidate set, and $\mathcal{C}_i$ denote the candidate set of $x_i$. Moreover, $\mathcal{C}_i$ consists of the top $k_2$ hottest nodes for $x_i$, i.e., the top $k_2$ nodes with the highest probability according to the generated heatmap $\boldsymbol{H}_i$. The initial tour is constructed by greedy selection. First, the start node $x_i$ is randomly selected from $\mathcal{X}_n$. Second, the hottest node in $\mathcal{C}_i$ that has not been traversed yet is selected as the next node. If all nodes in $\mathcal{C}_i$ have been traversed but the TSP tour is still incomplete, the nearest untraversed node is selected as the next node. The second step is conducted iteratively until all nodes are traversed.

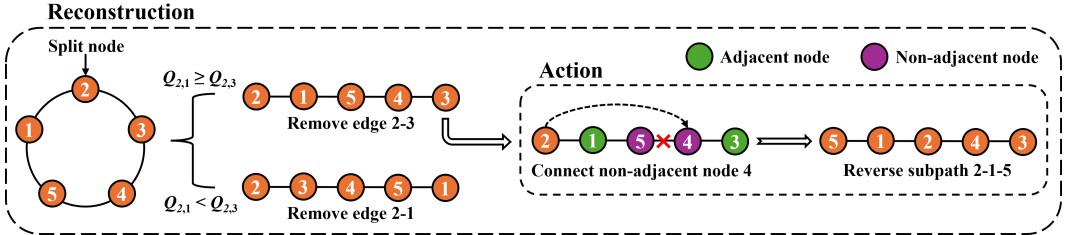

Figure 3: An example of reconstruction in RBS.

#### 4.3.2 RECONSTRUCTION

Reconstruction aims to enhance the search diversity and escape from local optima. First, a node is randomly selected as the split node. There are two edges directly connected to the split node in the tour, and the one with a smaller weight is selected and removed to form an acyclic path. Herein, the weights of edges are stored in $\boldsymbol{Q}$, where $\boldsymbol{Q}_{i,j}$ represents the importance of the edge $x_i$-$x_j$. $\boldsymbol{Q}$ is initialized as an all-zero symmetric matrix and is updated during the **Optimization and Edge Enhancement**. Then, taking the split node as the start node, a reconstruction action is conducted on the current acyclic path. In detail, a target node is selected from non-adjacent nodes of the start node based on the probability distribution $t_{i,j} = \frac{\boldsymbol{Q}_{i,j}+\varepsilon}{\sum_{x_h \in \mathcal{C}_i}(\boldsymbol{Q}_{i,h}+\varepsilon)}$, where $\varepsilon$ denotes a small number to prevent division by zero; $t_{i,j}$ represents the probability of $x_j$ being selected as the target node with $x_i$ as the start node. Finally, the acyclic path is updated by removing an edge of the target node and adding a new edge between the start node and the target node. Note that after a reconstruction action, the start node is also updated, and multiple reconstruction actions are conducted in a reconstruction process.

Take Figure 3 as an example to illustrate details. Initially, node 2 is the randomly selected split node. The two edges directly connected to node 2 are "2-1" and "2-3". If the weight of "2-3" is not larger than that of "2-1", i.e., $\boldsymbol{Q}_{2,1} \geq \boldsymbol{Q}_{2,3}$, "2-3" is removed from the tour and an acyclic path 2-1-5-4-3 is obtained. A reconstruction action is illustrated on the right side of Figure 3. Assuming node 4 is selected as the target node of the current start node (node 2), the edge between the target node and its previous adjacent node, i.e., the edge "5-4", is removed. A new edge between the start node and

the target node, i.e., the edge "2-4", is added. After that, we can obtain a new acyclic path 5-1-2-4-3 with node 5 as the start node for the next reconstruction action.

The reconstruction process iteratively conducts reconstruction actions and will terminate until any one of the following three criteria is met: (1) The tour formed by adding the last edge to the current acyclic path obtains improvement compared to the tour before reconstruction; (2) All non-adjacent nodes of the current start node have already been selected as target nodes; (3) The number of actions reaches the maximum threshold $M$.

### 4.3.3 OPTIMIZATION AND EDGE ENHANCEMENT

After the reconstruction, a tour is constructed by closing the reconstructed acyclic path. Subsequently, 2-Opt is employed to further optimize the tour, and the search space of 2-Opt is also confined to the candidate set of each node. In a 2-Opt swap, two selected edges $x_{i_1}$-$x_{j_1}$ and $x_{i_2}$-$x_{j_2}$ are transformed into $x_{i_1}$-$x_{j_2}$ and $x_{i_2}$-$x_{j_1}$. For every swap that obtains improvement to the tour, we update $\boldsymbol{Q}$ to enhance the weight of transformed edges as:

$$\boldsymbol{Q}_{i,j} \leftarrow \boldsymbol{Q}_{i,j} + \exp\big(-\frac{L(\Pi^{new})}{L(\Pi^{pre})}\big), \ \forall (i,j) \in \{(i_1, j_2), (i_2, j_1)\}, \tag{6}$$

where $\Pi^{pre}$ is the tour before the swap and $\Pi^{new}$ is the improved tour after the swap. The weight update will guide the next reconstruction, helping the reconstructed path escape local optima while preserving the optimal subpaths. Ultimately, it facilitates the attainment of a better tour in subsequent optimization.

## 5 EXPERIMENT

### 5.1 DATASETS

Let TSP-$n$ denote $n$-node TSP instances in uniform distribution. Our mixed-scale training set, abbreviated as TSP-Mix, is derived from a portion of the training data used by Joshi et al. (2019) and the detailed features are shown in Section 4.2. As a common setting, the test sets TSP-20/50/100/200/500/1K/10K are taken from part of those used by Fu et al. (2021), while TSP-2K/5K are the same as those used by Pan et al. (2023). In detail, the test sets consist of 1024 instances each for TSP-20/50/100; 128 instances each for TSP-200/500/1K; and 16 instances each for TSP-2K/5K/10K. For real-world instances, 78 instances with the number of nodes below 20K are selected from TSPLIB (Reinelt, 1991).

### 5.2 TRAINING

In RsGCN, we set the number of graph convolutional layers $l = 6$, the feature dimension $h = 128$, and the maximum number of adjacent nodes $k_1 = \min(50, n)$ for $n$-node instances. The Adam optimizer (Kingma & Ba, 2015) with a cosine annealing scheduler (Loshchilov & Hutter, 2017) is employed for training on TSP-Mix. We set $epochs = 3$ and $batch\ size = 32$, with the learning rate decaying cosinely from 5e-4 to 0. All experiments are conducted on an **NVIDIA H20** (96 GB) GPU and an **AMD EPYC 9654** (96-Core @ 2.40GHz) CPU. In our configuration, RsGCN only requires approximately 10 minutes per training epoch.

### 5.3 BASELINES

Our method is compared with: (1) **Classical Solvers/Algorithms:** Concorde (Applegate et al., 2006), LKH-3 (Helsgaun, 2017), 2-Opt (Croes, 1958); (2) **Transformer-Based Methods:** H-TSP (Pan et al., 2023), LEHD (Luo et al., 2023), GLOP (Ye et al., 2024), DRHG (Li et al., 2025); (3) **GCN-Based Methods:** Att-GCN (Fu et al., 2021), SoftDist (Xia et al., 2024), DIFUSCO (Sun & Yang, 2023), Fast T2T (Li et al., 2024).

All baselines are evaluated on the same test sets and hardware platform as ours. We set $trials = 10$K and enable $special$ option for LKH-3; limit the runtime to a reasonable maximum for Concorde to prevent unacceptable runtime when solving large-scale TSPs. For fairness in dealing with uniform-distribution instances, we strive to control the total runtime of neural methods to be equivalent to or longer than ours. Additional reproduction details on baselines are provided in Appendix C.

## 5.4 TESTING

For RBS, we set the maximum number of threads to 128 to fully utilize the CPU, set the runtime limit to $0.05n$ seconds, and randomly sample the maximum number of reconstruction actions $M \in [10, \min(40, n))$ for $n$-node instances. To account for distributional differences, we set the candidate size $k_2$ to 5 and 10 for uniform-distribution and real-world instances, respectively.

Three metrics are employed for performance evaluation, including (1) the average TSP tour length; (2) the average optimality gap; (3) the total solution time across all instances for each scale. Herein, the optimality gap of an instance is calculated as $(L/\hat{L}) - 1$, where $\hat{L}$ denotes the optimal tour length solved by Concorde. We conduct repeated experiments with 5 random seeds, reporting the average metrics for uniform-distribution instances and the best metrics for real-world instances in TSPLIB.

## 5.5 MAIN RESULTS

Tables 1 and 2 demonstrate RsGCN's strong generalization and state-of-the-art performance on both uniform-distribution and real-world TSPs. With only 3 epochs of training on TSP-Mix containing instances with up to 100 nodes, the two-stage framework RsGCN + RBS can generalize to 10K-node instances. RBS can also obtain competitive results without the guidance of RsGCN, validating its efficiency. In addition, Figure 4 shows that RsGCN requires the fewest learnable parameters among neural baselines, and RsGCN also requires the fewest training epochs, as shown in Appendix F.

Table 1: Comparison results on uniform-distribution TSPs. **RBS** here employs the 5-nearest neighbors as candidates, substituting for 5-hottest neighbors generated by RsGCN to comparatively evaluate RsGCN's guidance effect on RBS. The two-stage solution time is connected with +. The first-stage times of Att-GCN and DIFUSCO are taken from their publications due to reproducibility issues. Except for Concorde and LKH-3, we mark the smallest and second smallest gaps in red and blue, respectively.

| Method | TSP-20 | | | TSP-50 | | | TSP-100 | | | TSP-200 | | | TSP-500 | | |
|---|---|---|---|---|---|---|---|---|---|---|---|---|---|---|---|
| | Length | Gap(%) | Time | Length | Gap(%) | Time | Length | Gap(%) | Time | Length | Gap(%) | Time | Length | Gap(%) | Time |
| Concorde | 3.8403 | 0.0000 | 1.19s | 5.6870 | 0.0000 | 3.35s | 7.7560 | 0.0000 | 18.96s | 10.7191 | 0.0000 | 7.44s | 16.5464 | 0.0000 | 31.04s |
| LKH-3 | 3.8403 | 0.0000 | 28.32s | 5.6870 | 0.0000 | 56.52s | 7.7561 | 0.0003 | 55.33s | 10.7193 | 0.0022 | 7.41s | 16.5529 | 0.0391 | 10.12s |
| 2-Opt | 3.8459 | 0.1452 | 0.04s | 5.7812 | 1.6490 | 0.07s | 8.0124 | 3.3033 | 0.15s | 11.2332 | 4.7946 | 0.08s | 17.5688 | 6.1776 | 0.03s |
| H-TSP | — | — | — | — | — | — | — | — | — | 10.8276 | 1.0122 | 6.09s | 17.5962 | 6.3446 | 17.82s |
| LEHD | 3.8403 | 0.0000 | 10.00s | 5.6870 | 0.0005 | 25.00s | 7.7572 | 0.0147 | 45.00s | 10.7309 | 0.1103 | 15.00s | 16.6603 | 0.6881 | 54.00s |
| GLOP | 3.8406 | 0.0078 | 14.07s | 5.6935 | 0.1143 | 29.42s | 7.7839 | 0.3597 | 1.65m | 10.7952 | 0.7099 | 19.86s | 16.9089 | 2.1907 | 46.61s |
| DRHG | 3.8403 | 0.0000 | 10.00s | 5.6872 | 0.0039 | 25.00s | 7.7581 | 0.0272 | 45.00s | 10.7556 | 0.3410 | 15.00s | 16.6448 | 0.5947 | 50.00s |
| Att-GCN + MCTS | 3.8403 | 0.0000 | 2.39s + 10.14s | 5.6875 | 0.0090 | 15.91s + 23.16s | 7.7571 | 0.0137 | 24.21s + 46.13s | 10.7625 | 0.4035 | 20.62s + 12.99s | 16.7996 | 1.5308 | 31.17s + 51.53s |
| SoftDist + MCTS | 3.8745 | 0.8801 | 0.12s + 9.82s | 5.7692 | 1.4495 | 0.12s + 23.93s | 7.9553 | 2.5842 | 0.12s + 45.89s | 10.8971 | 1.6647 | 0.12s + 13.39s | 16.8161 | 1.6295 | 0.12s + 51.53s |
| DIFUSCO + MCTS | — | — | — | — | — | — | — | — | — | — | — | — | 16.6378 | 0.5519 | 3.61m + 51.40s |
| Fast T2T + 2-Opt | 3.8416 | 0.0348 | 53.00s | 5.6876 | 0.0115 | 1.00m | 7.7587 | 0.0337 | 1.23m | 10.7445 | 0.2364 | 28.00s | 16.6461 | 0.6024 | 37.00s |
| RBS | 3.8403 | 0.0000 | 10.31s | 5.6874 | 0.0066 | 22.40s | 7.7626 | 0.0854 | 42.14s | 10.7473 | 0.2634 | 12.46s | 16.6309 | 0.5108 | 27.15s |
| RsGCN + RBS | 3.8303 | 0.0000 | 0.15s + 9.17s | 5.6870 | 0.0000 | 0.20s + 20.18s | 7.7570 | 0.0120 | 0.41s + 40.20s | 10.7295 | 0.0966 | 0.12s + 10.07s | 16.5941 | 0.2884 | 0.36s + 25.09s |

| Method | TSP-1K | | | TSP-2K | | | TSP-5K | | | TSP-10K | | |
|---|---|---|---|---|---|---|---|---|---|---|---|---|
| | Length | Gap(%) | Time | Length | Gap(%) | Time | Length | Gap(%) | Time | Length | Gap(%) | Time |
| Concorde | 23.1218 | 0.0000 | 1.75m | 32.4932 | 0.0000 | 1.98m | 51.0595 | 0.0000 | 5.00m | 71.9782 | 0.0000 | 10.33m |
| LKH-3 | 23.1684 | 0.2017 | 17.96s | 32.6478 | 0.4756 | 29.75s | 51.4540 | 0.7727 | 2.06m | 72.7601 | 1.0864 | 7.06m |
| 2-Opt | 24.7392 | 6.9961 | 0.12s | 34.9866 | 7.6729 | 0.49s | 55.1501 | 8.0115 | 7.87s | 77.8939 | 8.2188 | 51.88s |
| H-TSP | 24.6679 | 6.6868 | 36.97s | 34.8984 | 7.4022 | 9.42s | 55.0178 | 7.7523 | 20.08s | 77.7446 | 8.0113 | 38.57s |
| LEHD | 23.6907 | 2.4603 | 1.48m | 34.2119 | 5.2892 | 2.15m | 59.6522 | 16.8290 | 9.59m | 90.5505 | 25.8026 | 54.45m |
| GLOP | 23.8377 | 3.0962 | 1.51m | 33.6595 | 3.5893 | 1.80m | 53.1567 | 4.1073 | 4.56m | 75.0439 | 4.2592 | 9.01m |
| DRHG | 23.3217 | 0.8648 | 1.50m | 32.8920 | 1.2274 | 2.01m | 51.6350 | 1.1271 | 5.00m | 72.8973 | 1.2770 | 9.00m |
| Att-GCN + MCTS | 23.6454 | 2.2650 | 43.94s + 1.71m | — | — | — | — | — | — | 74.3267 | 3.2627 | 4.16m + 17.38m |
| SoftDist + MCTS | 23.6622 | 2.3378 | 0.14s + 1.81m | 33.3675 | 2.6909 | 0.12s + 3.39m | 52.6930 | 3.1994 | 0.12s + 8.66m | 74.1944 | 3.0791 | 0.14s + 17.03m |
| DIFUSCO + MCTS | 23.4243 | 1.3083 | 0.12s + 1.71m | — | — | — | — | — | — | 73.8913 | 2.6579 | 0.14s + 17.55m |
| Fast T2T + 2-Opt | 23.3122 | 0.8236 | 2.95m | 32.7940 | 0.9258 | 2.03m | 51.7409 | 1.3345 | 4.35m | 72.9450 | 1.3432 | 17.00m |
| RBS | 23.2756 | 0.6657 | 52.38s | 32.7452 | 0.7757 | 1.68m | 51.5199 | 0.9018 | 4.18m | 72.8742 | 1.2448 | 8.37m |
| RsGCN + RBS | 23.2210 | 0.4293 | 0.94s + 50.40s | 32.6784 | 0.5698 | 0.49s + 1.68m | 51.3987 | 0.6645 | 2.83s + 4.17m | 72.6335 | 0.9103 | 11.83s + 8.36m |

Table 2: Average gaps(%) across different size intervals on TSPLIB instances. The results of other baselines are taken from previous works (Li et al., 2024; 2025). Configuration settings are provided below solvers in the header. Complete results on each TSPLIB instance can be found in Appendix D.

| Size | DIFUSCO $T_s$=50 | T2T $T_s$=50, $T_g$=30 | Fast T2T $T_s$=10, $T_g$=10 | BQ bs16 | LEHD RRC1K | GLOP more rev. | DRHG T=1K | RBS $k_2$=10 | RsGCN $k_1$=50, $k_2$=10 |
|---|---|---|---|---|---|---|---|---|---|
| <100 | 0.73 | 0.11 | 0.00 | 0.49 | 0.48 | 0.54 | 0.48 | 0.00 | 0.00 |
| [100, 200) | 0.91 | 0.39 | 0.25 | 1.66 | 0.20 | 0.79 | 0.15 | 0.37 | 0.00 |
| [200, 500) | 2.48 | 1.42 | 0.91 | 1.41 | 0.38 | 1.87 | 0.36 | 0.90 | 0.04 |
| [500, 1K) | 3.71 | 1.78 | 1.19 | 2.20 | 1.21 | 3.28 | 0.26 | 0.25 | 0.13 |
| ≥1K | — | — | — | 6.68 | 4.14 | 7.23 | 2.09 | 1.13 | 0.77 |
| All | — | — | — | 2.95 | 1.59 | 3.58 | 0.95 | 0.72 | 0.30 |

## 5.6 ABLATION STUDY

To validate the effectiveness of the subgraph-based rescaling, we set the following ablation variants: (1) **Rs×2:** applying subgraph-based rescaling, i.e., RsGCN; (2) **Rs×1:** constructing subgraphs without rescaling edge lengths; (3) **Rs×0:** disabling subgraph-based rescaling, equivalent to the vanilla Gated GCN; (4) **5-NN:** RBS with the 5-nearest neighbors as candidates. Figure 5 presents the gaps of each variant with respect to **5-NN**, revealing that the subgraph-based rescaling is crucial for improving GCNs' generalization. Without rescaling edges, **Rs×1** shows inferior generalization compared to **Rs×2**. Even worse, **Rs×0** fails to provide effective guidance for TSPs with more than 500 nodes. Figure 6 shows that the overall convergence speed follows **Rs×2 > Rs×1 > 5-NN > Rs×0**, validating the effective guidance of RsGCN (the y-axis represents the average tour length across 16 instances for each scale, and the x-axis represents the runtime). Furthermore, additional ablation study on RBS is presented in Appendix E.

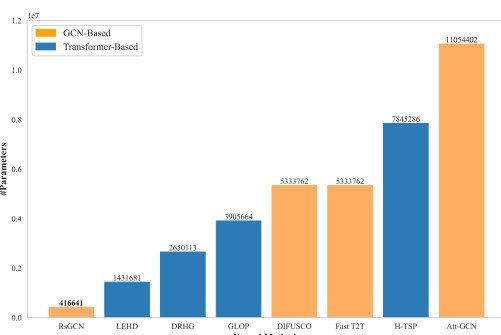

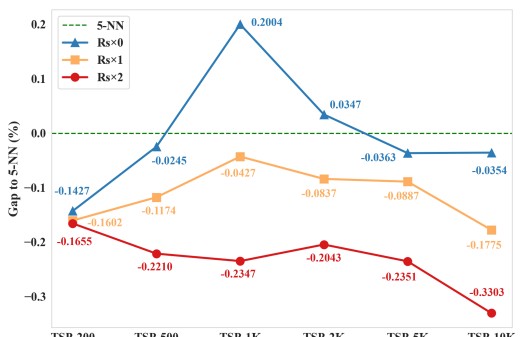

Figure 4: Comparison of neural model sizes.

Figure 5: Comparison of guidance effect among ablation variants.

Two metrics are incorporated to evaluate heatmaps on unseen-scale TSPs: (1) **Average Rank:** The average rank of optimal successors' heats, and (2) **Missing Rate in Top-5:** The probability that optimal successors are **not** included in the 5-hottest neighbors. Figure 7 presents the results of these two metrics, where **Dist$^{-1}$** denotes heatmaps obtained by calculating the reciprocal of distance matrices. Figure 7 indicates that **Rs×2** achieves superior overall performance, further validating the effectiveness of the subgraph-based rescaling. Furthermore, to visually assess heatmaps' quality, we introduce **ordered heatmaps** that are sorted according to the optimal tour. The detailed definition of the ordered heatmaps, along with the ablation results, is provided in Appendix A.

## 5.7 COMPUTATIONAL COST

Using publicly released data from previous studies, we compare the training cost of our RsGCN with Fast T2T (Li et al., 2024), as presented in Table 3. Fast T2T employs a Diffusion model as its decoder, which incurs additional training overhead. Moreover, it does not perform edge normalization, resulting in poor cross-scale generalization and requiring additional fine-tuning to generalize to large-scale TSPs. In contrast, our RsGCN achieves strong generalization while remaining lightweight, significantly reducing the training cost. Furthermore, Table 4 reports the single-batch

inference time and memory usage of RsGCN across different problem scales. RsGCN exhibits good linear memory scalability, suggesting its potential to address ultra-large-scale TSPs. In addition, more comparisons of training configurations are presented in Appendix F.

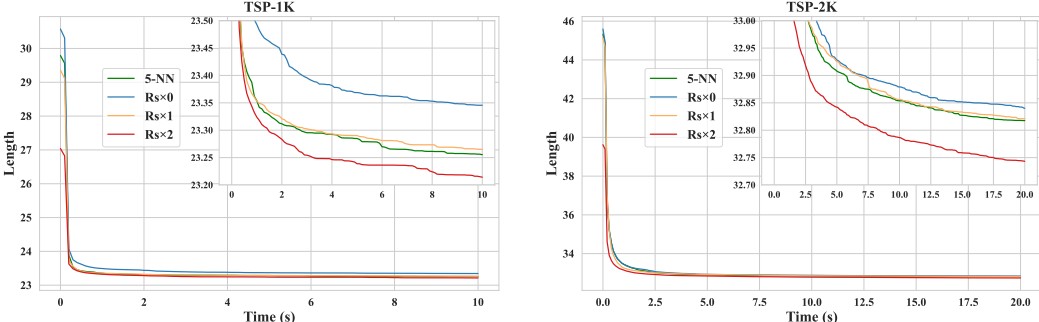

Figure 6: Comparison of boost to RBS's Convergence speed on TSP-1K/2K.

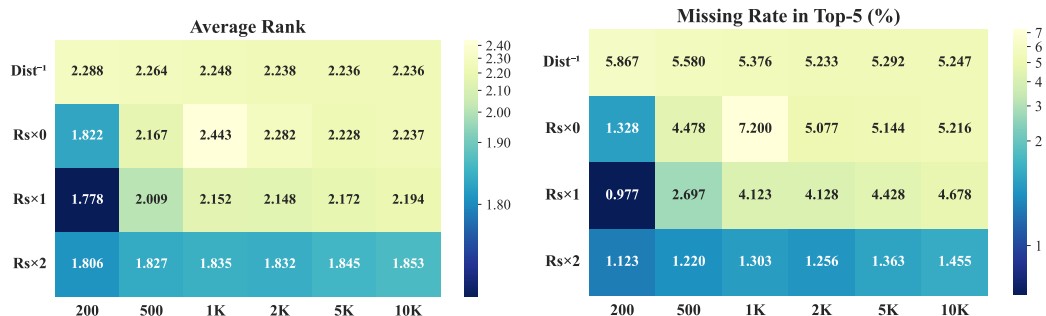

Figure 7: Two metrics that reveal the quality of heatmaps (smaller values are preferred).

Table 3: Comparison of training cost between Fast T2T and RsGCN.

| Model | Problem Scale | Dataset Size | Batch Size | GPU | Memory | Time |
|---|---|---|---|---|---|---|
| Fast T2T | TSP-50 | 1502K | 32 | $1 \times$ A100 | 16.5GB | 112h 45m |
| | TSP-100 | 1502K | 32 | | 23.2GB | 488h 12m |
| RsGCN | TSP-Mix {20,30,50,100} | 1000K | 32 | $1 \times$ A100 $1 \times$ H20 | 4.0GB 3.4GB | 50m 28m |

Table 4: Time and memory usage during inference.

| TSP- | 50 | 100 | 500 | 1K | 5K | 10K | 50K | 100K | 500K |
|---|---|---|---|---|---|---|---|---|---|
| **Time (s)** | 0.01 | 0.01 | 0.01 | 0.01 | 0.01 | 0.01 | 0.11 | 0.32 | 2.01 |
| **Memory (GB)** | 0.02 | 0.02 | 0.08 | 0.15 | 0.74 | 1.46 | 7.28 | 14.55 | 72.70 |

## 6 CONCLUSION

To enhance the cross-scale generalization capability of GCNs for solving TSPs, we propose a new RsGCN that incorporates the subgraph-based rescaling. The subgraph-based rescaling includes constructing subgraphs and rescaling subgraph edges to help RsGCN learn universal patterns across various scales. By 3-epoch training on a mixed-scale dataset composed of instances with up to 100 nodes, RsGCN can generalize to 10K-node instances without any fine-tuning. Extensive experimental results on uniform-distribution instances of 9 different scales from 20 to 10K and 78 real-world instances from TSPLIB validate the state-of-the-art performance of our method. In addition, the training cost and the number of learnable parameters are both significantly lower than those of the neural baselines.

# 7 ETHICS STATEMENT

This research adheres to established ethical standards. All datasets employed are either synthetic or publicly available, containing no sensitive or personally identifiable information. The study does not involve human participants and poses no privacy or security risks. All methods and experiments were conducted in accordance with relevant laws and accepted research integrity practices.

# 8 REPRODUCIBILITY STATEMENT

We have made efforts to ensure that the results reported in this paper are reproducible. The experimental setup, including datasets and model architectures, is detailed in Section 5 and Appendix B. The source code will be released publicly upon acceptance.

# 9 LLM USAGE STATEMENT

In this work, large language models (LLMs) were employed solely as tools to assist with writing, editing, and LaTeX formatting. Furthermore, the use of LLMs was limited to improving clarity, grammar, and presentation, and did not influence any scientific decisions, data processing, or interpretation of results.

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

## A    Ordered Heatmaps

Let $\hat{\boldsymbol{H}} \in \mathbb{R}^{n \times n}$ denote the ordered heatmap for an $n$-node instances. The procedure to obtain an ordered heatmap is described in Algorithm 1. An example of ideal ordered heatmap is shown in Figure 8, in which the neighboring grids along the main diagonal, as well as the grids in the lower-left and upper-right corners of the heatmap, have the largest heat values of 1 while the other grids have the heat value of 0. With such a transformation, we can intuitively investigate the quality of heatmaps by observing the distribution of heat value, i.e., a heatmap similar to the ideal heatmap is preferred.

Figure 9 displays the identical 100×100 local regions of the ordered heatmaps generated by **Rs×2/×1/×0** on two instances of TSP-500 and TSP-2K, respectively. Some cells close to the main diagonal in **Rs×0**' heatmap do not obtain heats, which easily lead to local optima. While for **Rs×1**, although the coverage around the main diagonal is enhanced compared to **Rs×0**, it suffers from the excessive dispersion of heats, which provides only limited guidance for the post-search. Overall, **Rs×2**'s heatmaps reveal a concentration of heat distribution around the main diagonal, demonstrating its effectiveness and superiority.

---

**Algorithm 1:** Obtain Ordered Heatmap

---

**Input:** The original heatmap $\boldsymbol{H}$, the optimal tour $\hat{\Pi}$
**Output:** Ordered heatmap $\hat{\boldsymbol{H}}$
**for** $i = 1$ *to* $n$ **do**
    $k \leftarrow$ index of $i$ in $\hat{\Pi}$;
    **for** $j = 1$ *to* $n$ **do**
        $h = j + k - 1$;
        **if** $h > n$ **then**
            $h = h - n$;
        $\hat{\boldsymbol{H}}_{i,j} = \boldsymbol{H}_{i,\pi_h}$;

---

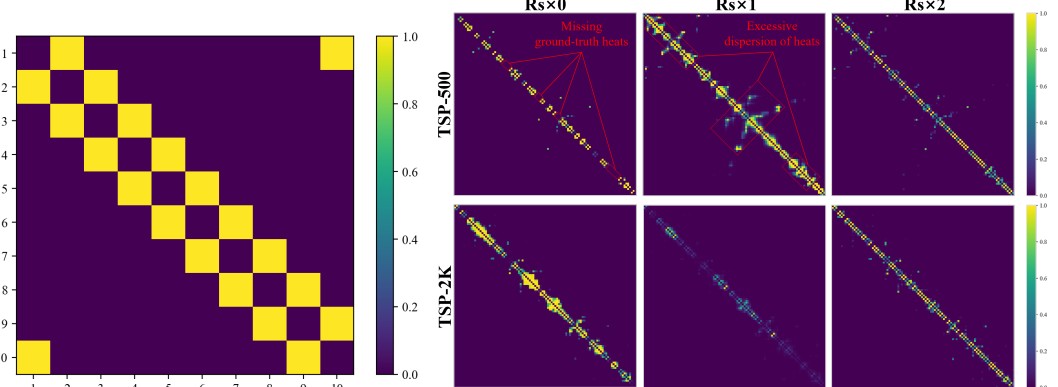

Figure 8: The ideal ordered heatmap of TSP-10.

Figure 9: Comparison of ordered heatmaps.

## B    Architecture of GCN

Our model adopts Residual Gated Graph Convnets (Bresson & Laurent, 2018; Joshi et al., 2019).

**Embedding Layer**    According to the problem definition, $x_i$ denotes the 2-dimensional coordinates in $\mathcal{X}_n$. Let $e_{i,j}$ denote the **rescaled** distance between $x_i$ and $x_j$ in subgraph $\mathcal{N}_i^{k_1}$. Subsequently, the global coordinate $x_i$ and the subgraph edge $e_{i,j}$ are linearly embedded to $h$-dimensional features as

follows:

$$x_i^0 = \boldsymbol{W}_1 x_i + b_1 \tag{7}$$

$$e_{i,j}^0 = \boldsymbol{W}_2 e_{i,j} + b_2 \tag{8}$$

where $\boldsymbol{W}_1 \in \mathbb{R}^{2 \times h}$ and $\boldsymbol{W}_2 \in \mathbb{R}^{1 \times h}$ are learnable parameter matrices, $b_1$ and $b_2$ are learnable biases. Such an embedding design retains the global distribution view while maintaining a unified subgraph perspective during GCN message aggregation.

**Graph Convolutional Layer** Given the embedded features $x_i^0$ and $e_{i,j}^0$, the graph convolutional node features $x_i^\ell$ and edge features $e_{i,j}^\ell$ at layer $\ell$ ($\geq 1$) are defined as follows:

$$x_i^\ell = x_i^{\ell-1} + \text{GELU}(\text{LN}(\boldsymbol{W}_3^\ell x_i^{\ell-1} + \sum_{x_j \in \mathcal{N}_i^{k_1}} \sigma(e_{i,j}^{\ell-1}) \odot \boldsymbol{W}_4^\ell x_j^{\ell-1})) \tag{9}$$

$$e_{i,j}^\ell = e_{i,j}^{\ell-1} + \text{GELU}(\text{LN}(\boldsymbol{W}_5^\ell e_{i,j}^{\ell-1} + \boldsymbol{W}_6^\ell x_i^{\ell-1} + \boldsymbol{W}_7^\ell x_j^{\ell-1})) \tag{10}$$

where $\boldsymbol{W}_{3 \sim 7} \in \mathbb{R}^{h \times h}$ are four learnable parameter matrices. LN denotes Layer Normalization (Ba et al., 2016), and GELU denotes the Gaussian Error Linear Unit (Hendrycks & Gimpel, 2016) used as the feature activation function. $\sigma$ denotes the sigmoid used as the gating activation function. $\odot$ denotes the Hadamard product operation.

**Projection Layer** Let $e_{i,j}^l$ denote the edge features in the last layer, where $l$ is the maximum number of graph convolutional layers. $e_{i,j}^l$ is projected to $\boldsymbol{H}_{i,j} \in (0, 1)$ as follows:

$$\boldsymbol{H}_{i,j} = \begin{cases} \sigma(\text{MLP}(e_{i,j}^l)), & \text{if } x_j \in \mathcal{N}_i^{k_1} \text{ and } i \neq j \\ 0, & \text{others} \end{cases} \tag{11}$$

where MLP is a two-layer perceptron with GELU as the activation function. $\boldsymbol{H}_{i,j}$ represents the probability that edge $x_i \to x_j$ exists in the predicted TSP tour.

## C    REPRODUCTION DETAILS ON BASELINES

For neural methods, we set the maximum batch size within the GPU memory limits to fully utilize the GPU. We set the maximum number of threads to 128 for CPU-dependent computation solvers Concorde, LKH-3, 2-Opt, MCTS, and RBS. For fairness, we adjust the hyperparameters to keep the runtime of baselines approximately the same. However, it remains difficult to ensure complete fairness, as the primary computational resources differ among solvers. Transformer-based methods leverage the GPU's powerful parallel processing capability to perform larger batch inference (typically covering all instances of each scale in one batch), whereas GCN-based methods mainly employ the GPU when generating heatmaps, with the post-search algorithms MCTS and RBS relying entirely on the CPU.

Among neural methods, H-TSP, DRHG, DIFUSCO, and Fast T2T employ fine-tuning strategies. We conduct evaluation using the fine-tuned DRHG for TSP-1K+. H-TSP, DIFUSCO, and Fast T2T are all fine-tuned for multiple specific instance scales. For these three solvers, if no fine-tuned model corresponding to the test scale is available, we use model weights with the fine-tuned scale closest to the test scale for evaluation.

## D    ADDITIONAL RESULTS ON TSPLIB

Note that we increase $k_2$ to 10 for TSPLIB, because RsGCN is trained only on uniform-distribution TSPs, making cross-distribution generalization to real-world TSPs challenging. We conduct an ablation study on the effect of $k_2$ on solving TSPLIB instances, reporting the minimum and average gaps over 5 repeated trials. As shown in Table 5, for the minimum gaps, the differences among $k_2 \in \{5, 7, 10\}$ are minor, while for the average gaps, $k_2 = 10$ performs better. Overall, for instances from distributions that RsGCN has not been trained on, appropriately enlarging the RBS candidate set improves search robustness.

Table 5: The effect of $k_2$ on solving TSPLIB instances.

| Size | Min Gap(%) | | | Avg Gap(%) | | |
|------|------------|---|---|------------|---|---|
| | $k_2$=5 | $k_2$=7 | $k_2$=10 | $k_2$=5 | $k_2$=7 | $k_2$=10 |
| <100 | 0.00 | 0.00 | 0.00 | 0.00 | 0.00 | 0.00 |
| [100, 200) | 0.01 | 0.00 | 0.00 | 0.24 | 0.23 | 0.23 |
| [200, 500) | 0.21 | 0.03 | 0.04 | 1.08 | 0.67 | 0.21 |
| [500, 1K) | 0.11 | 0.07 | 0.13 | 0.73 | 0.44 | 0.26 |
| ≥1K | 0.64 | 0.69 | 0.77 | 1.30 | 1.20 | 1.13 |
| All | 0.29 | **0.27** | 0.30 | 0.83 | 0.68 | **0.55** |

Tables 6 and 7 present the specific optimality gaps on 78 TSPLIB instances, where the results of DIFUSCO, T2T, and Fast T2T are sourced from Li et al. (2024), while those of POMO, BQ, LEHD, and DRHG are extracted from Li et al. (2025). Due to the distinctive and unusual node distributions of instances fl1577 and fl3795, we activate all neighbors as candidate nodes in RBS' State Initialization only for solving fl1577 and fl3795. Consequently, the optimality gaps decrease as follows: For RBS, 19.53% → 0.71% on fl1577 and 15.23% → 1.40% on fl3795; For RsGCN, 8.23% → 0.37% on fl1577 and 7.45% → 0.97% on fl3795. Note that without such a small adjustment on these two instances, the average optimality gap across all 78 instances obtained by the proposed RsGCN will grow to 0.49, which is still the best compared to other neural methods.

Table 6: Optimality gaps(%) on 78 TSPLIB instances (OOM refers to GPU out-of-memory).

| Instance | DIFUSCO $T_s$=50 | T2T $T_s$=50, $T_g$=30 | Fast T2T $T_s$=10, $T_g$=10 | POMO aug×8 | BQ bs16 | LEHD RRC100 | DRHG T=1K | RBS $k_2$=10 | RsGCN $k_1$=50, $k_2$=10 |
|----------|---------|-----|---------|------|----|------|------|-----|-------|
| a280 | 1.39 | 1.39 | 0.10 | 12.62 | 0.39 | 0.30 | 0.34 | 0.00 | 0.00 |
| berlin52 | 0.00 | 0.00 | 0.00 | 0.04 | 0.03 | 0.03 | 0.03 | 0.00 | 0.00 |
| bier127 | 0.94 | 0.54 | 1.50 | 12.00 | 0.68 | 0.01 | 0.01 | 0.00 | 0.00 |
| brd14051 | — | — | — | OOM | OOM | OOM | 4.02 | 2.62 | 1.91 |
| ch130 | 0.29 | 0.06 | 0.00 | 0.16 | 0.13 | 0.01 | 0.01 | 0.00 | 0.00 |
| ch150 | 0.57 | 0.49 | 0.00 | 0.53 | 0.39 | 0.04 | 0.04 | 0.00 | 0.00 |
| d198 | 3.32 | 1.97 | 0.86 | 19.89 | 8.77 | 0.71 | 0.26 | 0.01 | 0.01 |
| d493 | 1.81 | 1.81 | 1.43 | 58.91 | 8.40 | 0.92 | 0.31 | 0.01 | 0.27 |
| d657 | 4.86 | 2.40 | 0.64 | 41.14 | 1.34 | 0.91 | 0.21 | 0.12 | 0.12 |
| d1291 | — | — | — | 77.24 | 5.97 | 2.71 | 2.09 | 1.74 | 0.14 |
| d1655 | — | — | — | 80.99 | 9.67 | 5.16 | 1.57 | 0.83 | 1.76 |
| d2103 | — | — | — | 75.22 | 15.36 | 1.22 | 1.82 | 0.51 | 0.17 |
| d15112 | — | — | — | OOM | OOM | OOM | 3.41 | 2.30 | 2.17 |
| d18512 | — | — | — | OOM | OOM | OOM | 3.63 | 2.28 | 2.12 |
| eil51 | 2.82 | 0.14 | 0.00 | 0.83 | 0.67 | 0.67 | 0.67 | 0.00 | 0.00 |
| eil76 | 0.34 | 0.00 | 0.00 | 1.18 | 1.24 | 1.18 | 1.18 | 0.00 | 0.00 |
| eil101 | 0.03 | 0.00 | 0.00 | 1.84 | 1.78 | 1.78 | 1.78 | 0.00 | 0.00 |
| fl417 | 3.30 | 3.30 | 2.01 | 18.51 | 5.11 | 2.87 | 0.49 | 0.00 | 0.00 |
| fl1400 | — | — | — | 47.36 | 11.60 | 3.45 | 1.43 | 1.67 | 0.21 |
| fl1577 | — | — | — | 71.17 | 14.63 | 3.71 | 3.08 | 0.71 | 0.37 |
| fl3795 | — | — | — | 126.86 | OOM | 7.96 | 4.61 | 1.40 | 0.97 |
| fnl4461 | — | — | — | OOM | OOM | 12.38 | 1.20 | 0.89 | 0.91 |
| gil262 | 2.18 | 0.96 | 0.18 | 2.99 | 0.72 | 0.33 | 0.33 | 0.08 | 0.04 |
| kroA100 | 0.10 | 0.00 | 0.00 | 1.58 | 0.02 | 0.02 | 0.02 | 0.00 | 0.00 |
| kroA150 | 0.34 | 0.14 | 0.00 | 1.01 | 0.01 | 0.00 | 0.00 | 0.00 | 0.00 |
| kroA200 | 2.28 | 0.57 | 0.49 | 2.93 | 0.50 | 0.00 | 0.00 | 0.00 | 0.00 |
| kroB100 | 2.29 | 0.74 | 0.65 | 0.93 | 0.01 | -0.01 | -0.01 | 0.00 | 0.00 |
| kroB150 | 0.30 | 0.00 | 0.07 | 2.10 | -0.01 | -0.01 | -0.01 | 0.00 | 0.00 |
| kroB200 | 2.35 | 0.92 | 2.50 | 2.04 | 0.22 | 0.01 | 0.01 | 0.00 | 0.00 |
| kroC100 | 0.00 | 0.00 | 0.00 | 0.20 | 0.01 | 0.01 | 0.01 | 0.00 | 0.00 |
| kroD100 | 0.07 | 0.00 | 0.00 | 0.80 | 0.00 | 0.00 | 0.00 | 0.00 | 0.00 |
| kroE100 | 3.83 | 0.27 | 0.00 | 1.31 | 0.07 | 0.00 | 0.17 | 0.00 | 0.00 |
| lin105 | 0.00 | 0.00 | 0.00 | 1.31 | 0.03 | 0.03 | 0.03 | 0.00 | 0.00 |
| lin318 | 2.95 | 1.73 | 1.21 | 10.29 | 0.35 | 0.30 | 0.30 | 0.32 | 0.00 |
| linhp318 | 2.17 | 1.11 | 0.78 | 12.11 | 2.01 | 1.74 | 1.69 | 1.98 | 1.65 |
| nrw1379 | — | — | — | 41.52 | 3.34 | 8.78 | 1.41 | 0.42 | 0.58 |
| p654 | 7.49 | 1.19 | 1.67 | 25.58 | 4.44 | 2.00 | 0.03 | 0.00 | 0.00 |
| pcb442 | 2.59 | 1.70 | 0.61 | 18.64 | 0.95 | 0.04 | 0.27 | 0.35 | 0.00 |
| pcb1173 | — | — | — | 45.85 | 3.95 | 3.40 | 0.39 | 0.83 | 0.56 |
| pcb3038 | — | — | — | 63.82 | OOM | 7.23 | 1.01 | 0.69 | 0.74 |
| pr76 | 1.12 | 0.40 | 0.00 | 0.14 | 0.00 | 0.00 | 0.00 | 0.00 | 0.00 |
| pr107 | 0.91 | 0.61 | 0.62 | 0.90 | 13.94 | 0.00 | 0.00 | 0.00 | 0.00 |
| pr124 | 1.02 | 0.60 | 0.08 | 0.37 | 0.08 | 0.00 | 0.00 | 0.00 | 0.00 |
| pr136 | 0.19 | 0.10 | 0.01 | 0.87 | 0.00 | 0.00 | 0.00 | 0.00 | 0.00 |

Table 7: Optimality gaps(%) for 78 TSPLIB instances (continued from Table 6).

| Instance | DIFUSCO $T_s$=50 | T2T $T_s$=50, $T_g$=30 | Fast T2T $T_s$=10, $T_g$=10 | POMO aug×8 | BQ bs16 | LEHD RRC100 | DRHG T=1K | RBS $k_2$=10 | RsGCN $k_1$=50, $k_2$=10 |
|---|---|---|---|---|---|---|---|---|---|
| pr144 | 0.80 | 0.50 | 0.39 | 1.40 | 0.19 | 0.09 | 0.00 | 7.28 | 0.00 |
| pr152 | 1.69 | 0.83 | 0.19 | 0.99 | 8.21 | 0.27 | 0.19 | 0.18 | 0.00 |
| pr226 | 4.22 | 0.84 | 0.34 | 4.46 | 0.13 | 0.01 | 0.01 | 12.12 | 0.00 |
| pr264 | 0.92 | 0.92 | 0.73 | 13.72 | 0.27 | 0.01 | 0.01 | 0.00 | 0.00 |
| pr299 | 1.46 | 1.46 | 1.40 | 14.71 | 1.62 | 0.10 | 0.02 | 0.00 | 0.00 |
| pr439 | 2.73 | 1.63 | 0.50 | 21.55 | 2.01 | 0.33 | 0.12 | 0.98 | 0.03 |
| pr1002 | — | — | — | 43.93 | 2.94 | 0.77 | 0.67 | 0.37 | 0.17 |
| pr2392 | — | — | — | 69.78 | 7.72 | 5.31 | 0.56 | 0.32 | 0.56 |
| rat99 | 0.09 | 0.09 | 0.00 | 1.90 | 0.68 | 0.68 | 0.68 | 0.00 | 0.00 |
| rat195 | 1.48 | 1.27 | 0.79 | 8.15 | 0.60 | 0.61 | 0.57 | 0.22 | 0.00 |
| rat575 | 2.32 | 1.29 | 1.43 | 25.52 | 0.84 | 1.01 | 0.36 | 0.32 | 0.21 |
| rat783 | 3.04 | 1.88 | 1.03 | 33.54 | 2.91 | 1.28 | 0.47 | 0.42 | 0.22 |
| rd100 | 0.08 | 0.00 | 0.00 | 0.01 | 0.01 | 0.01 | 0.01 | 0.00 | 0.00 |
| rd400 | 1.18 | 0.44 | 0.08 | 13.97 | 0.32 | 0.02 | 0.36 | 0.02 | 0.00 |
| rl1304 | — | — | — | 67.70 | 5.07 | 1.96 | 0.79 | 0.56 | 0.42 |
| rl1323 | — | — | — | 68.69 | 4.41 | 1.71 | 1.26 | 0.20 | 0.24 |
| rl1889 | — | — | — | 80.00 | 7.90 | 2.90 | 0.95 | 1.26 | 0.14 |
| rl5915 | — | — | — | OOM | OOM | 11.21 | 1.97 | 1.31 | 0.82 |
| rl5934 | — | — | — | OOM | OOM | 11.11 | 2.69 | 2.87 | 1.42 |
| rl11849 | — | — | — | OOM | OOM | 21.43 | 3.94 | 3.11 | 1.79 |
| st70 | 0.00 | 0.00 | 0.00 | 0.31 | 0.31 | 0.31 | 0.31 | 0.00 | 0.00 |
| ts225 | 4.95 | 2.24 | 1.37 | 4.72 | 0.00 | 0.00 | 0.00 | 0.00 | 0.00 |
| tsp225 | 3.25 | 1.69 | 0.81 | 6.72 | -0.43 | -1.46 | -1.46 | -1.40 | -1.40 |
| u159 | 0.82 | 0.00 | 0.00 | 0.95 | -0.01 | -0.01 | -0.01 | 0.00 | 0.00 |
| u574 | 2.50 | 1.85 | 0.94 | 30.83 | 2.09 | 0.69 | 0.24 | 0.20 | 0.00 |
| u724 | 2.05 | 2.05 | 1.41 | 31.66 | 1.57 | 0.76 | 0.27 | 0.44 | 0.21 |
| u1060 | — | — | — | 53.50 | 7.04 | 2.80 | 0.48 | 0.35 | 0.39 |
| u1432 | — | — | — | 38.48 | 2.70 | 1.92 | 0.49 | 0.69 | 0.45 |
| u1817 | — | — | — | 70.51 | 6.12 | 4.15 | 2.05 | 0.74 | 0.46 |
| u2152 | — | — | — | 74.08 | 5.20 | 4.90 | 2.24 | 0.86 | 0.56 |
| u2319 | — | — | — | 26.43 | 1.33 | 1.99 | 0.30 | 0.50 | 0.53 |
| usa13509 | — | — | — | OOM | OOM | 34.65 | 11.82 | 2.17 | 1.48 |
| vm1084 | — | — | — | 48.15 | 5.93 | 2.17 | 0.14 | 0.25 | 0.16 |
| vm1748 | — | — | — | 62.05 | 6.04 | 2.61 | 0.54 | 0.31 | 0.17 |
| Average | — | — | — | 26.41 | 2.95 | 1.59 | 0.95 | 0.72 | 0.30 |

# E  ADDITIONAL ABLATION STUDY ON RBS

As described in Section 4.3.3, edge enhancement guides RBS toward more effective reconstructions, allowing it to overcome local optima. This strategy, which utilizes historical experience, has been used in both Monte Carlo search trees and ant colony algorithms. To validate the effect of edge enhancement, we conduct repeated experiments on large-scale TSPs with it turned off. As shown in Table 8, edge enhancement significantly improves the efficiency of RBS.

Table 8: The effect of edge enhancement in RBS on large-scale TSPs, where **E** and **w/o E** denote with and without edge enhancement, respectively.

| Type | TSP-2K | | TSP-5K | | TSP-10K | |
|---|---|---|---|---|---|---|
| | Length | Gap(%) | Length | Gap(%) | Length | Gap(%) |
| **E** | 32.6784 | 0.5698 | 51.3987 | 0.6645 | 72.6335 | 0.9103 |
| **w/o E** | 32.7679 | 0.8455 | 51.5379 | 0.9370 | 72.7828 | 1.1178 |

Figure 10 shows the average optimal gaps (%) under different combinations of two hyperparameters: the maximum number of actions $M$ and candidate set sizes $k_2$. We select 16 instances for each scale. The runtime limit of RBS is set to $0.05n$ seconds for $n$-node instances. The x-axis represents $M$ with different sampling ranges $[10, 20]$, $[10, 30]$, $[10, 40]$, $[10, 50]$, and $[10, 60]$, while the y-axis represents $k_2$ with different values across $[4, 7]$. As shown in Figure 10, $k_2$ has a more significant impact on RBS. For larger instance scales, a smaller $k_2$ tends to help RBS achieve smaller optimality gaps. As a larger $k_2$ expands the search space, the efficiency of reconstruction is weakened and the runtime of each 2-Opt step is prolonged in dealing with large-scale instances.

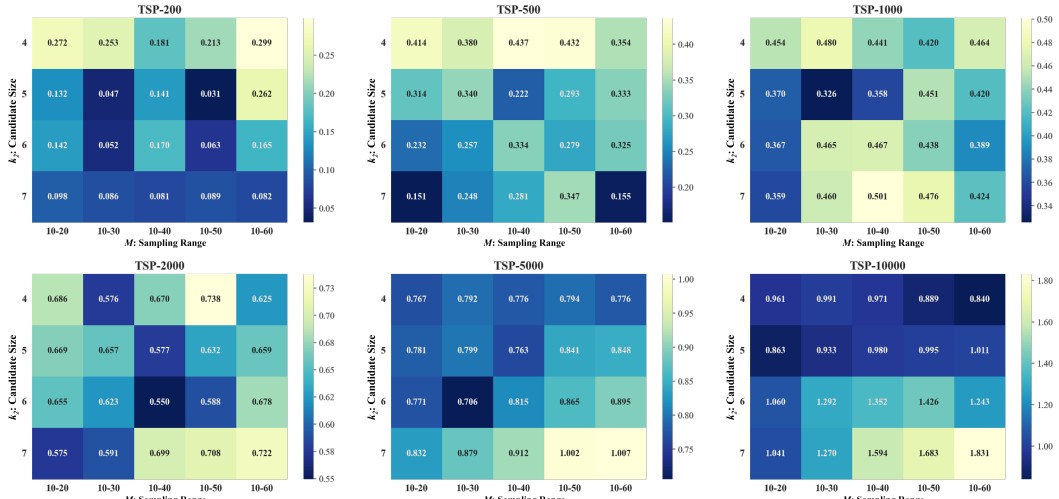

Figure 10: Ablation study on the hyperparameters $M$ and $k_2$ of RBS.

## F    COMPARISON OF TRAINING CONFIGURATIONS

Table 9 presents the configurations of the main training phase for each neural solver, where SL denotes supervised learning and RL denotes reinforcement learning. Our RsGCN requires significantly fewer learnable parameters and training epochs compared to other baselines, substantially reducing the training cost. Moreover, our main results in Section 5.5 also demonstrate that supervised learning on small-scale TSPs is an effective training strategy for RsGCN. In GLOP, "×3" indicates that GLOP requires combining 3 neural models with identical parameter sizes during a single solving process, whereas the "+" in H-TSP denotes the combination of two neural models with different parameter scales. Both GLOP and H-TSP employ reinforcement learning, requiring a very large number of training epochs.

Table 9: Comparison on training configurations of neural methods.

| Paradigm | Solver | #Parameters | Training Epochs |
|---|---|---|---|
| SL | RsGCN | 0.417M | 3 |
| | LEHD | 1.43M | 150 |
| | DRHG | 2.65M | 100 |
| | DIFUSCO | 5.33M | 50 |
| | Fast T2T | 5.33M | 50 |
| | Att-GCN | 11.1M | 15 |
| RL | GLOP | 1.30M × 3 | >500 |
| | H-TSP | 5.34M + 2.51M | >500 |

## G    VISUALIZATION OF SUBGRAPH-BASED RESCALING

Figure 11 visually illustrates the effect of the subgraph-based rescaling. As shown in the first row, the node distributions become denser in the unit square as the number of nodes increases. First, 5-NN selection is conducted, and each node forms a subgraph containing itself and its 5-nearest neighbors. Then, we observe that the magnitude of edge lengths in subgraphs varies across different scales. As shown in the second row, the average distances between the given node and its neighbors in the three instances are 0.1793, 0.0878, and 0.0206 respectively. Thus, we employ Uniform Unit Square Projection to rescale subgraph edges, obtaining the rescaled subgraph shown in the third row. We can see that the average distances are rescaled to the same magnitude across different scales, while the proportion of the subgraphs is preserved. Overall, the subgraph-based rescaling enhances the cross-scale generalization of RsGCN.

# H   LIMITATIONS AND BROADER IMPACTS

**Limitations**   Our current methodology focuses on solving Symmetric TSP. Future work requires further extensions to broader routing problems, including the Asymmetric TSP (ATSP) and Capacitated Vehicle Routing Problem (CVRP). Our proposed methods still trails behind Concorde and LKH-3. However, Concorde and LKH-3 highly rely on expert-crafted heuristics specifically designed for TSPs.

**Broader Impacts**   Our work highlights the importance of subgraph-based rescaling and provides insightful implications for future research to improve the generalization of neural combinatorial optimization (NCO) solvers. Furthermore, our solver's small parameter scale and low training cost will prompt the community to rethink the practical utility and cost-effectiveness of NCO solvers.

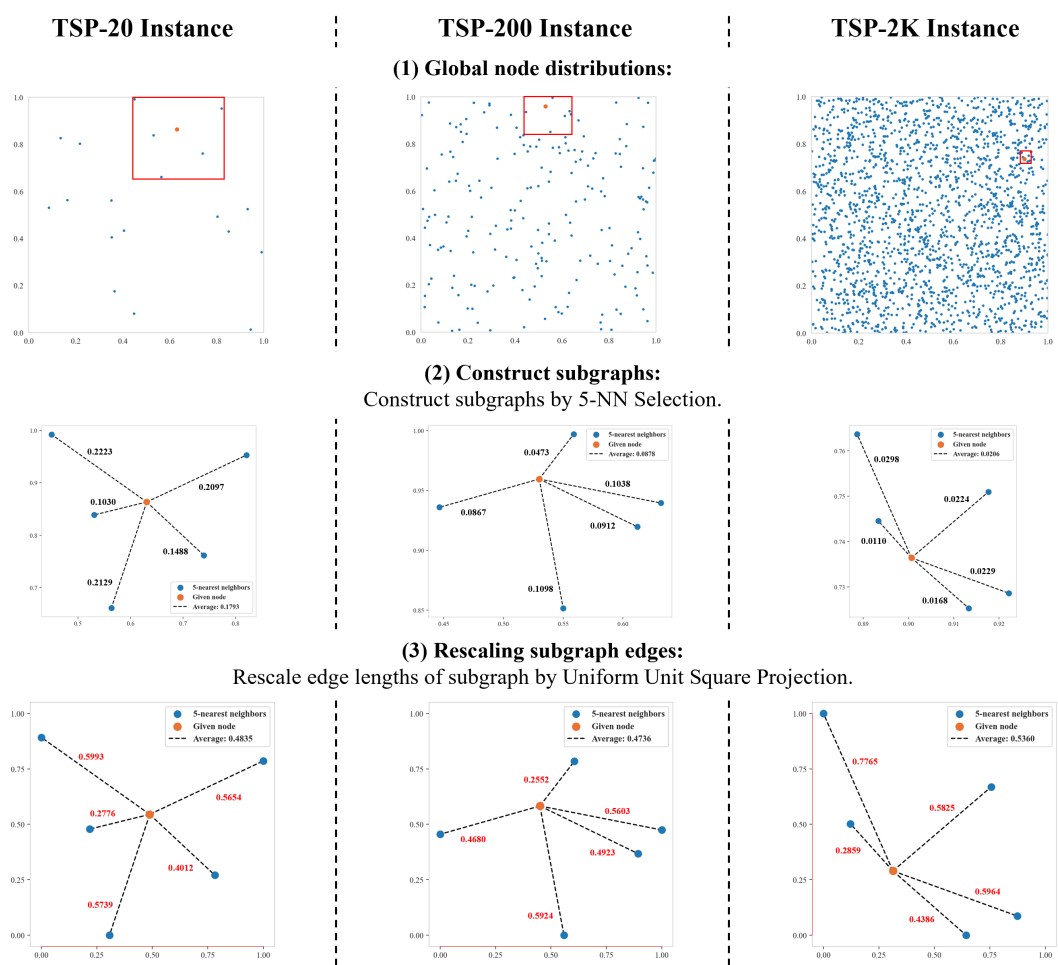

Figure 11: Visualization of the subgraph-based rescaling on TSP-20/200/2K.

