# OpenReview forum: "RsGCN: Subgraph-Based Rescaling Enhances Generalization of GCNs for Solving Traveling Salesman Problems"
_ICLR.cc/2026/Conference — Submitted to ICLR 2026_

### Official Review · Reviewer_R5e5 · 2025-10-30

**Soundness:** 3
**Presentation:** 2
**Contribution:** 2
**Rating:** 4
**Confidence:** 5

**Summary:**

This paper introduces RsGCN, a new Graph Convolutional Network (GCN) model designed to enhance generalization in solving Traveling Salesman Problems (TSP) across varying problem scales. The authors address the issue of poor cross-scale generalization in GCN-based TSP solvers by incorporating a subgraph-based rescaling technique. This approach involves constructing subgraphs using k-Nearest Neighbors (k-NN) and rescaling subgraph edges using Uniform Unit Square Projection to standardize the edge lengths. RsGCN is combined with a Reconstruction-Based Search (RBS) post-search strategy, which further refines the solution by enhancing the diversity of the search space and avoiding local optima. The model demonstrates significant improvements in both performance and training efficiency, achieving state-of-the-art results across TSP instances ranging from 20 to 10,000 nodes while requiring fewer parameters and training epochs compared to previous methods.

**Strengths:**

1.The introduction of subgraph-based rescaling to adaptively normalize edge lengths is a novel idea that addresses the key challenge of generalizing across different TSP scales. This approach allows the model to learn universal patterns from smaller TSP instances, which can then be successfully applied to larger-scale problems without fine-tuning.
2.The proposed method achieves impressive results with only 3 epochs of training on a mixed-scale dataset containing instances up to 100 nodes. This level of efficiency, combined with the ability to generalize to 10K-node instances, is a significant strength, as it reduces training time and resources compared to other neural methods.
3.The experimental results demonstrate the superiority of RsGCN + RBS over other methods, including both classical solvers (Concorde, LKH-3) and recent neural network-based approaches. The model’s generalization ability is highlighted by its performance across various test sets, including uniform-distribution instances and real-world TSP instances from TSPLIB.

**Weaknesses:**

1.While RsGCN handles large-scale problems better than previous methods, the model still requires full passes for each decoding step, which might be computationally intensive for extremely large instances (e.g., 100K nodes). Optimizations in terms of decoding efficiency and inference time could be explored.


2.The experiments primarily focus on uniform-distribution instances, and the paper acknowledges that RsGCN has not been specifically trained on non-uniform distributions. Although the model generalizes well to real-world instances, a more detailed exploration of how RsGCN adapts to non-uniform distributions (e.g., city clustering) would be valuable.


3.The choice of k-NN and k2 for the RBS algorithm is critical, and while the ablation studies show solid results, the sensitivity of the model to these parameters could be more clearly quantified. A more detailed analysis of how varying these parameters impacts model performance would strengthen the study.

**Questions:**

1.While the model performs well on instances with up to 10,000 nodes, how does RsGCN perform on even larger-scale instances (e.g., 100K nodes)? Are there any optimizations or trade-offs that can be made to further improve scalability?
2.Since RsGCN was trained primarily on uniform-distribution datasets, how well does it generalize to non-uniform TSP distributions (e.g., cities with clustered or irregular distances)? Can the model be easily adapted to handle real-world TSP instances with more complex structures?
3.The paper introduces RBS, but how does this algorithm compare to other common post-search methods, such as Monte Carlo Tree Search (MCTS) or 2-Opt in terms of performance, flexibility, and computational cost? Is RBS consistently superior in all cases, or could another post-search method be more effective for specific TSP instances?
4.The paper mentions the selection of k1 and k2 as hyperparameters in the subgraph and RBS design. How sensitive is the performance of RsGCN to changes in these parameters, and can automated methods be used to select the optimal hyperparameters for different TSP instance scales?

---

> ### Author Response · Authors · 2025-11-20
>
> ## **Response to Question 1**
>
> We additionally evaluate the scalability of RsGCN on ultra-large TSP instances, using 16 TSP-50K and 16 TSP-100K test cases. To keep the overall runtime reasonable, we disable the `SUBGRADIENT` option in LKH-3 to avoid excessive preprocessing time. The maximum runtime of both LKH-3 and RBS is set to $0.035n$ seconds, where $n$ denotes the number of nodes. The candidate set sizes for both RBS and LKH-3 are fixed to $5$, and the maximum number of reconstruction actions $M$ in RBS is uniformly sampled from $[10, 60]$. The detailed results are presented in the table below.
>
> As can be seen, RsGCN provides highly effective guidance for RBS, achieving significantly better solution quality compared to 5-NN + RBS. Moreover, RsGCN + RBS even outperforms LKH-3 on TSP-100K. Meanwhile, RsGCN requires only 8 seconds and 17 seconds on a GPU to infer on the 16 TSP-50K and 16 TSP-100K instances, respectively. These results demonstrate the strong cross-scale generalization and excellent scalability of RsGCN.
>
> |   Method    |   TSP-50K    |   TSP-100K   |
> | :---------: | :----------: | :----------: |
> |             |    Length    |    Length    |
> |    LKH-3    | **168.8568** |   288.6964   |
> | 5-NN + RBS  |   188.1163   |   438.1735   |
> | RsGCN + RBS |   169.7974   | **285.5128** |
>
> ------
>
> ## **Response to Question 2**
>
> Table 2 in the main text and Table 6 in the appendix report the performance of RsGCN on real-world TSP instances. The comprehensive results demonstrate that RsGCN, trained solely on uniform-distribution instances, still exhibits excellent generalization to real-world instances, including instances with clustered distributions.
>
> ------
>
> ## **Response to Question 3**
>
> In fact, RBS can be viewed as an improved and simplified version of MCTS. While traditional MCTS performs $k$-Opt optimization based on a Markov chain, RBS adopts a probabilistic multi-step reconstruction strategy followed by deterministic refinement using 2-Opt.
>
> The performance difference between MCTS and RBS can be observed in Table 1 of the main paper, where the rows `SoftDist + MCTS` and `RBS` correspond to the settings without neural network guidance. Clearly, RBS achieves better performance than MCTS on TSPs of 9 different scales.
>
> From an implementation perspective, RBS is also significantly simpler than MCTS. Moreover, RBS does not require the full heatmap as input, which avoids the substantial memory overhead of loading large heatmaps and improves both its flexibility and computational efficiency.
>
> In summary, based on the existing experimental evidence, RBS surpasses MCTS in terms of performance, efficiency, and flexibility.
>
> ------
>
> ## **Response to Question 4**
>
> Based on our empirical validation and observations from optimal solutions, a subgraph size of $k_1=50$ is already sufficient to robustly cover all promising neighbors. In general, $k_1$ does not require dedicated tuning. The parameter $k_2$ only affects the RBS search process. Appendix E presents and analyzes the joint impact of $M$ and $k_2$ on RBS. The choice of $k_2$ should take into account the quality of the candidate set. When the candidate set is of lower quality, $k_2$ should be increased to enlarge the search space. The parameter $M$ mainly controls the reconstruction range; if sufficient time is available, increasing $M$ may help escape local optima.
>
> For future improvements, designing an adaptive strategy for adjusting $M$ is promising. For example, one may initially sample $M$ from $[10,30]$. If no better solution is found within a certain time under this setting, the algorithm could automatically adjust $M$ to $[20,40]$, thereby expanding the reconstruction range and enabling the search to escape local optima.
>
> ------
>
> ## **Remarks**
>
> Thank you for your valuable and professional review comments. We will include the additional experiments in the next version of the paper. If you have any further questions, please feel free to let us know. We hope our responses address your concerns and contribute to an improved rating.

---

### Official Review · Reviewer_WuKt · 2025-10-31

**Soundness:** 3
**Presentation:** 4
**Contribution:** 3
**Rating:** 6
**Confidence:** 4

**Summary:**

This paper introduces RsGCN, a subgraph-based rescaling method to enhance the cross-scale generalization of GCNs for solving Traveling Salesman Problems (TSPs). The key innovation is subgraph construction via k-NN and edge-length rescaling using Uniform Unit Square Projection, normalizing node/edge distributions across scales to allow training on small instances (≤100 nodes) while generalizing to large instances (up to 10K nodes). Combined with a novel Reconstruction-Based Search (RBS) that uses adaptive-weight reconstruction to escape local optima, the framework achieves state-of-the-art results on 9 uniform-distribution scales and 78 real-world TSPLIB instances.

**Strengths:**

1. Subgraph-based edge normalization effectively eliminates scale-dependent distortions, enabling robust zero-shot generalization.
2. Training on mixed-scale instances (≤100 nodes) for only 3 epochs suffices for 10K-node generalization.
3. RsGCN uses the fewest parameters among neural baselines (416K parameters vs. 1.4M–11M) and scales linearly in inference.
4. Outperforms GCN/Transformer baselines on both synthetic and real-world TSPs.

**Weaknesses:**

1. The idea of using k-NN to prune a neural architecture is not really novel, although it is suitable to solving TSP problems.
2. The idea of reconstruction-based search is not really novel either, similar to the improvement-based solvers introduced before.

**Questions:**

1.  If the results of rescaling mechanism are processed by LKH3 for heuristic search, would it be helpful to LKH-3?
2. I am particularly curious about whether the author have any novel perspectives on neural network applications for TSP solving beyond the proposed improvement-based (or reconstruction-based) paradigm. How might these insights inform future enhancements to RSGCN?

---

> ### Author Response · Authors · 2025-11-19
>
> ## **Q1. Guiding LKH-3 by RsGCN**
>
> Through additional experiments, we find that the candidate set generated by RsGCN also provides efficient guidance for LKH's search. The efficiency of guidance is reflected in two aspects: ① shorter time for generating candidate sets and preprocessing, and ② obtaining better solutions with less solution time.
>
> The plain LKH employs the Ascent algorithm to construct candidate sets. The Ascent algorithm is relatively complex and time-consuming. Our RsGCN, relying on GPU acceleration, generates candidate sets much faster than the Ascent algorithm. Thus, using RsGCN to provide candidate sets can significantly reduce the runtime.  Our experiments validate that the candidate sets generated by RsGCN are of higher quality, which can not only reduce solution time but also obtain better solutions.
>
> In detail, we conducted experiments with `MAX_TRIALS` set to 10 and 1000, respectively. The number of repeated `RUNS` in the search phase is set to 10, while `MAX_CANDIDATES` is uniformly set to 5. Ascent-LKH and RsGCN-LKH denote the plain LKH and the LKH with RsGCN to construct candidate sets, respectively. The hyperparameter settings for LKH in Ascent-LKH and RsGCN-LKH are the same and follow the default setting. Particularly, we enable the `SPECIAL` option in LKH to avoid introducing excessive handcrafted heuristics, thereby highlighting the quality of the candidate set.
>
> The test set is the same as in the paper. We report three metrics as follows:
>
> 1.  **Preprocess Time:** The average time consumed for preprocessing operations. Specifically, the average inference time of RsGCN-LKH is included in the preprocessing time, which is separately shown in the left of "+".
> 2.  **Search Time:** The average time consumed for a search in LKH for each instance.
> 3.  **Tour Length:** The average tour length for each instance across 10 repeated experiments.
>
> Since the differences in results on smaller scales are minor, the experimental results on TSP-2K/5K/10K are shown in the table below. The experimental results demonstrate the considerable potential of the RsGCN-LKH framework, as well as the powerful generalization capability of RsGCN and the high quality of its generated candidate sets.
>
> |                   |                 | TSP-2K      |             |                 | TSP-5K      |             |                  | TSP-10K     |             |
> | ----------------- | --------------- | ----------- | ----------- | --------------- | ----------- | ----------- | ---------------- | ----------- | ----------- |
> | `MAX_TRIALS=10`   | Preprocess Time | Search Time | Tour Length | Preprocess Time | Search Time | Tour Length | Preprocess Time  | Search Time | Tour Length |
> | Ascent-LKH        | 4.21s           | **0.01s**   | 33.3138     | 28.48s          | **0.04s**   | 52.3988     | 137.23s          | **0.10s**   | 73.9893     |
> | RsGCN-LKH         | **0.03s+1.01s** | **0.01s**   | **33.2877** | **0.18s+8.13s** | **0.04s**   | **52.3605** | **0.74s+44.29s** | **0.10s**   | **73.8620** |
> |                   |                 | **TSP-2K**  |             |                 | **TSP-5K**  |             |                  | **TSP-10K** |             |
> | `MAX_TRIALS=1000` | Preprocess Time | Search Time | Tour Length | Preprocess Time | Search Time | Tour Length | Preprocess Time  | Search Time | Tour Length |
> | Ascent-LKH        | 4.22s           | **0.23s**   | **32.9324** | 29.24s          | **0.77s**   | 52.0166     | 140.05s          | **2.23s**   | 73.5852     |
> | RsGCN-LKH         | **0.03s+1.01s** | **0.23s**   | 32.9742     | **0.18s+8.09s** | 0.78s       | **51.9791** | **0.74s+45.13s** | 2.36s       | **73.4834** |
>
> ------
>
> ## **Q2. Insights into GCNs for TSP**
>
> As an **anisotropic** GCN, RsGCN has a natural advantage in handling **asymmetric** distance matrices, which enables it to effectively tackle the Asymmetric TSP (ATSP). In addition, existing GCN-based methods typically rely on **global message aggregation**, which inevitably leads to substantial memory consumption when scaling to large instances. By leveraging the autoregressive characteristics of Transformer-based methods, we believe that a subgraph-autoregressive GCN framework holds strong potential for addressing ATSP and ACVRP—with improved scalability and efficiency. This could be a promising direction for future work.
>
> ------
>
> ## **Remarks**
>
> Thank you for your valuable comments and for your interest in GCN-based methods. We hope that our responses have addressed your concerns and will lead to an improved evaluation of our work. If you have any further questions, please feel free to let us know.

---

### Official Review · Reviewer_S3J9 · 2025-10-31

**Soundness:** 2
**Presentation:** 2
**Contribution:** 2
**Rating:** 4
**Confidence:** 4

**Summary:**

This paper introduces ``RsGCN``, which normalizes edge distances at the node level to eliminate scale-sensitivity across instances of different sizes. Additionally, it proposes ``RBS``, a novel K-OPT algorithm based on ``MCTS``. After only three rounds of supervised training on small-scale TSP instances, the combined ``RsGCN+RBS`` framework generalizes to large-scale TSP problems and achieves good performance.

**Strengths:**

1. The proposed method incurs low training costs and, according to experimental results, delivers good performance and generalization.

2. The paper conducts a thorough ablation study on the use of edge normalization and node K-NN.

**Weaknesses:**

1. Clearly, the sole focus on TSP is a major limitation of this paper. The proposed ``RsGCN+RBS`` is an improved variant of the earlier ``Att-GCN+MCTS`` pipeline. The latter, over the past few years, has not demonstrated any clear trend toward being extended to other combinatorial problems.

2. The authors do not report the results right after ``STATE INITIALIZATION``. Similar to ``Att-GCN``, the performance gains may largely come from the post-processing steps. The underlying idea of these post-processing techniques is essentially derived from ``LKH``, i.e., iteratively applying K-OPT to improve solution quality. This is also why these approaches struggle to extend to problems where LKH performs poorly, such as the ``CVRP``.

**Questions:**

1. Scaling edges differently for each node preserves their relative ordering, yet because every node has its own scaling factor the transformed problem is no longer equivalent to the original one; indeed, the optimal tour may change. So ``RsGCN`` is only used to screen the candidate set for ``RBS``, rather than training the ``GCN`` to learn to predict the solution as in previous works?

2. Figures 5 and 7 partly demonstrate the performance gains of ``RsGCN``, but all these experiments are tightly coupled with ``RBS``. I understand the authors’ intention to treat ``RsGCN+RBS`` as an integrated whole; nevertheless, I would still like to see the results when ``RBS`` is completely removed, i.e., after only ``STATE INITIALIZATION``.

3. Unlike MCTS, RBS does not rely on a heatmap and can thus be regarded as a generic post-processing strategy. Could the authors supplement results for other methods combined with RBS, e.g., ``Fast-T2T + RBS`` and ``DIFUSCO + RBS``?

4. The authors mention that future work will extend to ``ATSP`` and ``CVRP``. I would like to know how the proposed ``RsGCN+RBS`` pipeline could be applied to these problems.

---

> ### Author Response · Authors · 2025-11-19
>
> ## **Response to Question 1**
>
> I understand your point. It is worth noting that, based on our empirical validation and observations from optimal solutions, a subgraph size of $k_1=50$ is already sufficient to robustly cover all promising neighbors. In addition, for the input of RsGCN, we use the original node coordinates together with the rescaled subgraph edge lengths, which allows the model to preserve both local features and global context.
>
> RsGCN can also directly predict solutions when combined with post-search methods such as greedy search or 2-Opt. The corresponding results are provided in **"Response to Question 2"**.
>
> ------
>
> ## **Response to Question 2**
>
> We would like to clarify that our purpose in introducing RBS is primarily to provide a fast proxy for evaluating heatmap quality. In general, higher-quality heatmaps tend to yield better efficiency for *any* post-search method.
>
> For further validation, we repeated the experiments with two basic and intuitive post-search methods, i.e., **greedy search (G)** and **greedy search + 2-Opt (G + 2-Opt)** . The greedy search corresponds to `STATE INITIALIZATION`. We adopt **Fast T2T**, one of the state-of-the-art GCN-based methods, as our baseline. For fairness and to properly assess generalization, we use the publicly available Fast T2T weights trained on TSP-100. In terms of post-search settings, greedy search samples only a single solution, and 2-Opt follows the Fast T2T implementation with the default maximum iteration limit of **5000**.
>
> The two tables shown below presents the experimental results with greedy search (G) and G + 2opt for post-search, respectively (Rs×1 is a RsGCN's variant that does not rescale subgraph edges). We can see that RsGCN continues to deliver superior performance on large-scale TSPs, whereas Fast T2T exhibits clear signs of overfitting. It is also worth noting that **RsGCN requires substantially fewer parameters and much lower training cost** compared to Fast T2T.
>
> | Method (G) |   TSP-1K |   TSP-2K  |   TSP-5K    |   TSP-10K   |
> | :-----: | :------: | :------: | :-------: | :-------: |
> |  Fast T2T  | 29.5189 | 46.8250 |   88.7764   |  144.7111   |
> |    Rs×1| 28.6332 | 39.9434 |   62.8221   |   88.2919   |
> |   RsGCN  | **28.2555** | **39.7594** | **61.9926** | **87.3582** |
>
> | Method (G + 2-Opt) | TSP-1K  | TSP-2K  | TSP-5K  | TSP-10K   |
> | :-----: | :-----: | :-----: | :-----: | :-----: |
> |   Fast T2T   |   24.3895 |34.9044|55.8094|   79.0679   |
> | Rs×1 | 24.2784 | 34.2103|53.5836|75.6381|
> |  RsGCN| **24.0825** | **33.8883** | **53.0748** | **74.7902** |
>
> ------
>
> ## **Response to Question 3**
>
> Due to the differences in heatmap organization between Fast T2T/DIFUSCO and our RsGCN, extracting candidate sets that are compatible with our RBS format from their source code is non-trivial. Therefore, we are unable to provide `Fast T2T+RBS` and `DIFUSCO+RBS` results at this moment.
>
> Nevertheless, the relative performance of `Fast T2T+RBS` can be reasonably inferred from the `Fast T2T+G (+2-Opt)` results shown above. Our experiments show that the performance comparison under `greedy search (+2-Opt)` is generally consistent with that under `RBS`, since this trend reflects the underlying quality of candidate sets.
>
> ------
>
> ## **Response to Question 4**
>
> Thanks for your interest. Below we provide a preliminary roadmap for extending our approach to **ATSP** and **CVRP**.
>
> - **ATSP:** Since RsGCN is an anisotropic GCN, it naturally handles asymmetric distance matrices and can therefore generate candidate sets for ATSP. To enable RBS on ATSP, we can apply the **JV Transformation** [1] to convert an ATSP instance with $n$ nodes into an equivalent TSP instance with $2n$ nodes, and then perform RBS on the transformed instance. This aligns ATSP with the TSP setting and allows RBS to be directly applied.
> - **CVRP:** A standard way to solve CVRP [2] is to decompose it into: (i) **node clustering**, and (ii) **intra-cluster route optimization**. RsGCN can be trained to focus on producing feasible clusters that satisfy CVRP constraints. Each cluster can then be treated as a TSP instance, upon which RBS is used to efficiently optimize the local tour. This framework can leverage the robust global representation of RsGCN while exploiting the efficiency of RBS as a heuristic search.
>
> ------
>
> ## **Remarks**
>
> Thank you for your professional and forward-looking comments. We will include the corresponding results and clarifications in the next version of the paper. If you have any further questions, please feel free to let us know. We hope that our responses can adequately address your concerns and contribute to an improved rating.
>
> -------
>
> ## **Reference**
> [1] Transforming asymmetric into symmetric traveling salesman problems. *Operations Research Letters*, 2(4):161–163, 1983.
>
> [2] Generalize learned heuristics to solve large-scale vehicle routing problems in real-time. *The Eleventh International Conference on Learning Representations*, 2023.

---

### Official Review · Reviewer_GShj · 2025-11-01

**Soundness:** 3
**Presentation:** 2
**Contribution:** 3
**Rating:** 6
**Confidence:** 4

**Summary:**

This paper addresses two critical and persistent challenges in neural combinatorial optimization (NCO) for the Traveling Salesman Problem (TSP): poor cross-scale generalization and high training costs. The authors identify that GCNs are highly sensitive to "scale-dependent features," particularly the distribution of edge lengths, which changes as problem instances grow larger even when normalized in a unit square. The paper proposes RsGCN and RBS to deal with the two challenges. Strong empirical results are presented.

**Strengths:**

- Simple but effective: The core idea of the RsGCN model is refreshingly straightforward. The authors correctly identify that as TSP instances grow, the nodes in the unit square get denser, and the average edge lengths shrink. GCNs, being sensitive to the distribution of their input features (edge weights), fail to generalize. The solution of subgraph-based rescaling isn't a massive new architecture, a complex attention mechanism, or a heavy generative decoder.

- Exceptional training efficiency: The model achieves SOTA performance after training for only 3 epochs on small instances , which the authors report takes ~30-50 minutes on an A100/H20 GPU. This is orders of magnitude faster than competing diffusion-based and RL methods (e.g. fast t2t) that require hundreds of hours of training and fine-tuning. Combined with its minimal parameter count (0.417M), this makes RsGCN a highly practical and scalable solution.

**Weaknesses:**

- Limited contributions on the scope of studied problems. Only tsp is evaluated. More problems e.g. cvrp should be included to demonstrate the effectiveness.


- RBS vs. Classical Heuristics: The RBS algorithm is a sophisticated, multi-stage local search (destroy, repair, 2-opt, adaptive weighting). It is a strong heuristic in its own right, which is good, but it also means the performance lift is not purely from the learned GCN. Table 1 shows that RBS (using 5-NN, no GCN) is already a top-tier solver. This is a common feature of NCO, but it's worth noting that a significant part of the performance gain comes from a well-engineered (non-learned) search heuristic.

**Questions:**

- how to set k the number of nearest neighbors for instances of different scales? Would a different k from training cases when applied on testing cases affect the performance?

---

> ### Author Response · Authors · 2025-11-18
>
> ## **Further Validation of RsGCN**
>
> In fact, RBS can be viewed as a simplified and lightweight variant of MCTS. We would like to clarify that our purpose in introducing RBS is primarily to provide a fast proxy for evaluating heatmap quality. In general, higher-quality heatmaps tend to yield better efficiency for *any* post-search method.
>
> For further validation, we repeated the experiments with two basic and intuitive post-search methods, i.e., **greedy search (G)** and **greedy search + 2-Opt (G + 2-Opt)** . We adopt **Fast T2T**, one of the state-of-the-art GCN-based methods, as our baseline. For fairness and to properly assess generalization, we use the publicly available Fast T2T weights trained on TSP-100. In terms of post-search settings, greedy search samples only a single solution, and 2-Opt follows the Fast T2T implementation with the default maximum iteration limit of **5000**.
>
> The two tables shown below presents the experimental results with greedy search (G) and G + 2opt for post-search, respectively (Rs×1 is a RsGCN's variant that does not rescale subgraph edges). We can see that RsGCN continues to deliver superior performance on large-scale TSPs, whereas Fast T2T exhibits clear signs of overfitting. It is also worth noting that **RsGCN requires substantially fewer parameters and much lower training cost** compared to Fast T2T.
>
> | Method (G) |   TSP-1K    |   TSP-2K    |   TSP-5K    |   TSP-10K   |
> | :--------: | :---------: | :---------: | :---------: | :---------: |
> |  Fast T2T  |   29.5189   |   46.8250   |   88.7764   |  144.7111   |
> |    Rs×1    |   28.6332   |   39.9434   |   62.8221   |   88.2919   |
> |   RsGCN    | **28.2555** | **39.7594** | **61.9926** | **87.3582** |
>
> | Method (G + 2-Opt) |   TSP-1K    |   TSP-2K    |   TSP-5K    |   TSP-10K   |
> | :----------------: | :---------: | :---------: | :---------: | :---------: |
> |      Fast T2T      |   24.3895   |   34.9044   |   55.8094   |   79.0679   |
> |        Rs×1        |   24.2784   |   34.2103   |   53.5836   |   75.6381   |
> |       RsGCN        | **24.0825** | **33.8883** | **53.0748** | **74.7902** |
>
> ------
>
> ## **Response to the Question**
>
> Based on extensive experiments and observations of optimal solutions, we find that setting the number of nearest neighbors $k$ to **30–50** is generally the best choice for symmetric TSPs. This range robustly covers all promising neighbors. You may also refer to *“Rethinking Neural Combinatorial Optimization for Vehicle Routing Problems with Different Constraint Tightness Degrees,” arXiv:2505.24627 (2025)*, which reports similar findings on CVRP.
>
> Regarding the training and test configurations, our experiments suggest that using consistent values of $k$ is important. Otherwise, a non-negligible performance drop may occur. We hypothesize that this degradation is caused by a shift in feature distributions, as inconsistent $k$ affects the stability of message aggregation during inference.
>
> ------
>
> ## **Remarks**
>
> Thank you for your constructive and professional review comments, which have been extremely helpful for our work. We will include the results based on greedy search and greedy search + 2-Opt in the subsequent version of the paper. If you have any further questions, please feel free to let us know. We hope that our responses have addressed your concerns and may contribute to an improvement in the rating.

---

### Meta-Review · Area_Chair_Xgda · 2026-01-02

**Summary:**

This paper introduces RsGCN, a subgraph-based rescaling method to enhance the cross-scale generalization of GCNs for solving Traveling Salesman Problems (TSPs). There are detailed analyses in this manuscript. The proposed RsGCN demonstrates clear generalization ability and efficiency in training. This paper received two positive evaluations and two negative ones. After comprehensive consideration, I have decided to reject this paper. Reasons are as follows:

1. Novelty: neither the idea of rescaling method nor reconstruction is new for TSP; the rescaling method is widely considered in GLOP [1], INVIT [2], et al., and the RBS method is committed by authors as a special case of MCTS.

2. Experimental limitation: the distance-based reduction (rescaling) methods can hardly be applied to CVRP because the depot should be specially considered. The experiment is limited to TSP, which is not enough.


[1] Ye, Haoran, et al. "Glop: Learning global partition and local construction for solving large-scale routing problems in real-time." Proceedings of the AAAI conference on artificial intelligence. Vol. 38. No. 18. 2024.

[2] Fang, Han, et al. "Invit: A generalizable routing problem solver with invariant nested view transformer." arXiv preprint arXiv:2402.02317 (2024).

I strongly recommend that the authors make significant revisions based on the suggestions the reviewers have given regarding experimental and methodological design, and submit the manuscript to a subsequent conference.

**Reviewer Concerns:**

I believe some concerns about detailed implementations may be solved.

**Reviewer Scores:**

I don't think reviewers with negative evaluations will defend this paper.

---

### Decision · Program_Chairs · 2026-01-26

Reject